# Valorization of Value-Added Resources from the Anaerobic Digestion of Swine-Raising Manure for Circular Economy in Taiwan

Yu-Ru Lee [1] and Wen-Tien Tsai [2,*]

1   Graduate Institute of Environmental Management, Tajen University, Pingtung 907, Taiwan; yuru@tajen.edu.tw
2   Graduate Institute of Bioresources, National Pingtung University of Science and Technology, Pingtung 912, Taiwan
*   Correspondence: wttsai@mail.npust.edu.tw; Tel.: +886-8-770-3202

**Abstract:** Due to the benefits of mitigating greenhouse gas emission and upgrading farmland fertilization, the valorization of liquor and biogas digestate from the anaerobic digestion of swine manure has attracted much attention in recent years. This article is based on the updated data/information from the official websites for summarizing the status of the swine-raising industry and innovative manure management, relevant sustainable development indicators, and inter-ministry promotion regulations in Taiwan. The survey findings revealed that the carbon dioxide emission reduction in 2019 was equivalent to about 36,000 metric tons based on a total of 2.35 million metric tons liquor and biogas digestate applied and 2 million swine heads for the biogas-to-power. Obviously, the regulatory measures by the Council of Agriculture, the Environmental Protection Administration, and the Ministry of Economic Affairs have provided economic and financial supports towards the reduction of $67.39 \times 10^3$ metric tons carbon dioxide equivalent by 2030. Using the principles of biorefinery and zero-waste, the integration of anaerobic digestion, by-products (liquor and digestate), and biogas-to-power for treating swine manure is a win-win-win option for environmental, energy, and economic benefits.

**Keywords:** swine manure; anaerobic digestion; biogas; digestate; sustainable development goal; regulatory promotion

## 1. Introduction

The valorization of waste biomass for the production of biogas in the anaerobic digestion process has attracted much attention in recent years [1]. More importantly, the purified biogas containing high purity of methane can be used for electricity generation, thus mitigating global warming due to its high global warming potential (i.e., 25, relative to carbon dioxide). On the other hand, biogas digestate is another value-added resource produced from the anaerobic digestion at the bottom because it contains valuable nutrients for plant growth and crop production as an organic fertilizer [2]. It should be noted that heavy metals may be found in biogas digestate and/or swine manure, causing it to be easily leached from soil of fertilized farmland [3–5]. In addition, the wastewater from the swine-raising farms without adequate biological treatment may cause adverse impacts on surface water quality and air quality by the eutrophication and the odor, respectively.

To reach environmental goals in connection with the livestock industry, such as improving river quality and circular economy objectives, Taiwan's Environmental Protection Administration (EPA), jointly ventured with the Council of Agriculture (COA), announced the amendments to the "Water Pollution Control Measures and Test Reporting Management Regulations" under the authorization

of the Water Pollution Control Act on 24 November 2015. A featured chapter on the reuse of liquor and biogas digestate from the anaerobic digestion of swine manure for farmland fertilization has been added to the regulation. Over the period of 2017–2019, the goals for promoting an irrigation area have been achieved from 584 hectares in 2017 to 2645 hectares in 2019. In 2019, the carbon dioxide ($CO_2$) emission reduction was equivalent to about 36,000 metric tons based on a total of 2.35 million metric tons liquor and biogas digestate applied and 2 million swine heads for the biogas-to-power [6].

As compared to the traditional three-step system [7], the swine-raising wastewater directly treated by the anaerobic digestion not only creates biogas for power generation, but also produces the value-added resources (i.e., liquor and biogas digestate) for farmland fertilization [5,8]. In Taiwan, the central ministries, including the COA, the EPA, and the Ministry of Economic Affairs (MOEA), have provided economic and financial supports under the regulatory promotion. In line with the sustainable development goals (SDGs) launched by the Taiwanese government in 2018, the reuse of liquor and digestate from anaerobic digestion (AD) as fertilizers for agricultural lands or organic agriculture has been listed as one target in the 12th SDGs. The objectives of this paper were to summarize the status of the swine-raising industry and innovative manure management, as well as relevant sustainable development indicators in Taiwan. Furthermore, the inter-ministry regulations for promoting the value-added resources from the anaerobic digestion of manure were addressed in the paper to gain multiple benefits of environmental protection, green energy, and agricultural production, thus creating a circular economy in the swine-raising industry. Therefore, this case report will be based on the updated data/information from the official websites of the COA, the EPA, and the MOEA.

## 2. Status of the Swine-Raising Industry and Anaerobic Treatment of Manure for Recycling in Taiwan

### 2.1. Current Status of the Swine-Raising Industry in Taiwan

According to the official survey by the COA, swine farms raised about 5514 thousand heads by the end of 2019, as listed in Table 1 [9]. During the past fifteen years, total heads on swine-raising farms indicated a decreasing trend from 6.8 million heads in 2004 to about 5.5 million heads in recent years (2014–2019). This trend could be ascribed to the global competition due to Taiwan entering the World Trade Organization (WTO) on 1 January 2002. In addition, the environmental protection authorities also established strict effluent standards for the swine-raising industry to prevent excrement and manure from deteriorating the environmental quality under the legal systems. Therefore, many small-scaled swine-raising farms have been driven out of industry [10]. Table 1 also lists the statistics of total heads on farms, showing that total swine-raising farms of over 5000 heads were 139 farms with 1,274,042 heads in 2019. It should be noted that the economic scale for installing a biogas-to-power system has been considered to be over 5000 heads on swine-raising farms. In this regard, about 23.1% of total swine heads in Taiwan can be promoted for installing the AD process, thus producing the value-added resources like biogas and digestate.

**Table 1.** Statistics of swine heads on farms classified by scale in Taiwan [a].

| Scale (Heads) | Farms | Percentage (%) | Heads on Farms | Percentage (%) |
|---|---|---|---|---|
| 1~99 | 1955 | 29.4% | 58,506 | 1.1% |
| 100~199 | 650 | 9.8% | 98,374 | 1.8% |
| 200~299 | 382 | 5.7% | 94,338 | 1.7% |
| 300~499 | 614 | 9.2% | 245,770 | 4.5% |
| 500~999 | 1453 | 21.9% | 1,103,261 | 20.0% |
| 1000~1999 | 1054 | 15.9% | 1,475,149 | 26.7% |
| 2000~2999 | 248 | 3.7% | 596,831 | 10.8% |
| 3000~4999 | 150 | 2.3% | 567,940 | 10.3% |
| 5000 or more | 139 | 2.1% | 1,274,042 | 23.1% |
| Sum | 6645 | 100.0% | 5,514,211 | 100.0% |

[a] Source [9]; statistics to the end of November 2019.

### 2.2. Current Status of Swine Manure Treatment in Taiwan

Because swine excrement and manure contain lots of organic matter and dissolved nutrients, the wastewater from swine-raising farms before discharging into water bodies must be treated to avoid serious pollution of receiving streams. In the past, the three-step wastewater treatment system (i.e., liquid–solid separation, anaerobic digestion, and aerobic treatment) has been successfully adopted in Taiwan for additional biogas-to-power generation [5,7,11]. In most cases, the discharge of treated swine-raising wastewater can meet the effluent standards, which will be described thereafter. Currently, the central competent authorities (i.e., the EPA and the COA) have jointly ventured the applications of swine farms to utilize the liquor and digestate from the AD process as organic fertilizers in the nearby farmlands since 2016. Figure 1 shows the innovative process of valorizing the value-added resources from the swine-raising manure management in Taiwan and also summarized its energy and environmental benefits. Under the authorization of the Water Pollution Control Act, the EPA announced the amendments to the "Water Pollution Control Measures and Test Reporting Management Regulations", adding a new chapter regarding AD liquor and digestate as fertilizers for farmlands. The plan of valorizing AD-based liquor and digestate as fertilizers for farmlands must be submitted to the COA for application and review. Upon approval, the permit should be further reported to the local competent authorities in the implementation stage. The information about the plan of valorizing AD-based liquor and digestate as fertilizers will be subsequently addressed in detail. In addition, the liquor treated by the AD process can be reused for irrigation of plantation land if it meets the effluent standards and was approved by the local competent authorities. If the digestate does not meet the requirements for an organic fertilizer, it will be handled by the waste management regulations.

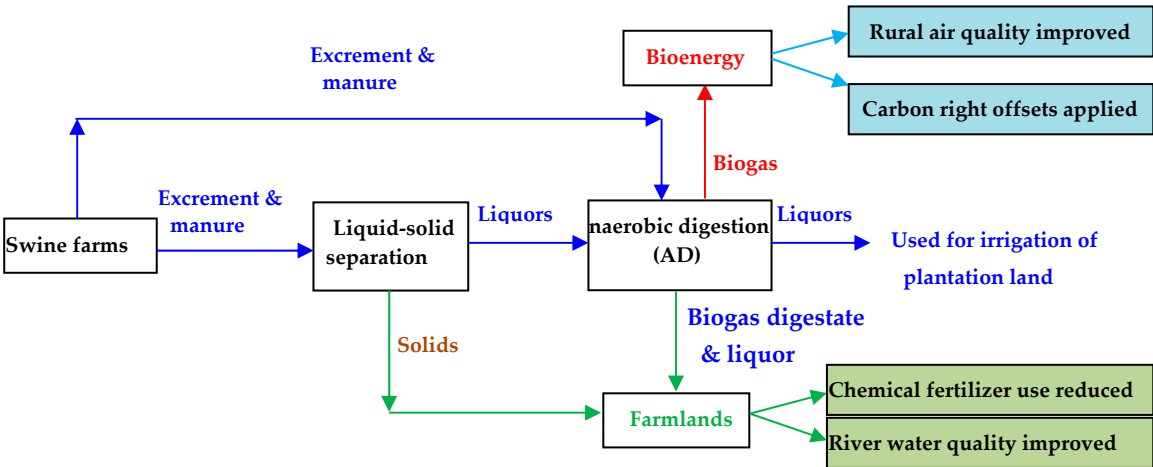

**Figure 1.** Flow-sheet of valorizing the value-added resources from the swine-raising manure management in Taiwan and its energy and environmental benefits.

### 2.3. Taiwan's Sustainable Development Goals with Relevance to the Livestock Industry

In 2015, the United Nations (UN) declared the SDGs in the 2030 Agenda for Sustainable Development [12], including those interrelated to poverty, inequality, climate change, environmental degradation, peace, and justice. They are the blueprint to achieve a better and more sustainable future for all by 2030. Based on the 17 items in the SDGs, the government launched Taiwan's SDGs in 2018, which comprises 18 goals, 143 targets, and 342 indicators. In Taiwan's SDGs, building a nuclear-free homeland by 2025 was listed as 18th SDG. Among these SDGs in Taiwan, the Target 12.8 was to promote environment-friendly and circular agriculture for the purposes of reducing the soil and water pollution due to agricultural applications and residues. Furthermore, the Indicator 12.8.2 aimed at the biogas production by valorizing manure with $375 \times 10^4$ swine heads by 2020, and the reduction of

$67.39 \times 10^3$ metric tons carbon dioxide ($CO_2$) equivalent by 2030. Table 2 shows the status of swine manure recycling (2017–2019) and its SDGs in Taiwan by 2030 [6].

**Table 2.** Status of swine manure recycling and its sustainable development goals (SDGs) in Taiwan.

| Indicators/Goals | Status | | | SDGs by 2030 | |
|---|---|---|---|---|---|
| | **2017** | **2018** | **2019** | | |
| Irrigation amount (metric ton) | $65.4 \times 10^4$ | $229 \times 10^4$ | $235 \times 10^4$ | - | Biogas production by valorizing manure with $375 \times 10^4$ heads swine |
| Irrigation area (hectare) | 584 | 1952 | 2645 | - | Reduction of $67.39 \times 10^3$ metric tons carbon dioxide ($CO_2$) equivalent |
| Accumulative head swine | $34 \times 10^4$ | $145 \times 10^4$ | $200 \times 10^4$ | - | $250 \times 10^4$ metric tons of irrigation amount maintained |

## 3. Regulations for Promoting Value-Added Resources from the Anaerobic Digestion of Swine Manure

As mentioned above, the issues for the valorization of value-added resources (i.e., biogas digestate and biogas-to-power) from the anaerobic fermentation of swine manure are related to the central competent authorities, including the COA, the EPA, and the MOEA. Using AD liquor and solid digestate as agricultural fertilizer and other reuse purposes, instead of flushing livestock slurry away, reduces water and electricity costs for the enterprises concerned. Livestock slurry is also diverted from polluting receiving bodies in Taiwan's rivers. The information about the regulations for promoting the value-added resources from swine manure will be addressed in the following sections by using the official regulatory website [13].

### 3.1. Council of Agriculture (COA)

#### 3.1.1. Fertilizer Management Act

In Taiwan, the Fertilizer Management Act (FMC) was promulgated on 18 June 1999 and subsequently revised on 19 June 2002. Its legislative purposes were to ensure soil fertility, promote agricultural productivity, and protect the farm environment. According to Article 4 of the Act, the central governing agency shall announce the categories, types, and specifications of fertilizers, including nitrogen fertilizer, phosphorus fertilizer, potassium fertilizer, minor trace elements fertilizer, organic matter fertilizer, compound fertilizer, plant growth aid, microbial fertilizer, and other fertilizers. Obviously, the biogas digestate belongs to one of the organic matter fertilizers due to its richness in soil nutrients [14–20]. Table 3 lists the composition specifications/limits of organic matter fertilizers (livestock manure compost item No. 5–09), including the specifications for the essential ingredients and the limits of toxic metals/elements.

**Table 3.** Composition specifications for organic fertilizers derived from livestock manure in Taiwan.

| Constituent | Specification (Maximum/Minimum Limit) |
|---|---|
| Essential ingredients | |
| Organic matter | $\geq 40\%$ |
| Total nitrogen | $\geq 1.0\%, \leq 4.0\%$ |
| Total phosphoric anhydride | $\geq 1.0\%, \leq 6.0\%$ |
| Total potassium oxide | $\geq 0.5\%, \leq 5.0\%$ |
| Toxic metals/elements | |
| Arsenic (As) | $\leq 25.0$ mg |
| Cadmium (Cd) | $\leq 2.0$ mg |
| Chromium (Cr) | $\leq 150$ mg |
| Copper (Cu) | $\leq 100$ mg |
| Mercury (Hg) | $\leq 1.0$ mg |
| Nickel (Ni) | $\leq 25.0$ mg |
| Lead (Pb) | $\leq 150$ mg |
| Zinc (Zn) | $\leq 500$ mg |

In addition, the livestock manure compost must meet the additional limitation requirements, which are listed below:

- The resulting compost fertilizer does not mix with chemical fertilizers, minerals, sludge, plant residues, fish powders, meat bone powders, kitchen waste, carbonized rice husk, peat, or residues treated by chemical methods;
- The resulting compost fertilizer does not mix with industrial waste;
- The moisture should be below 35.0%;
- The pH value should range from 5.0 to 9.0, which should be marked on the container;
- The ratio of carbon-to-nitrogen should range from 10.0 to 20.0.

### 3.1.2. Animal Industry Act

This act, recently revised on 24 November 2010, was to regulate and provide guidance to the livestock/poultry farming business for preventing environmental pollution and facilitating the development of the animal industry. Therefore, the farm owners must submit a plan for the environmental pollution prevention/control as a result of livestock excrement and manure. This plan must be reviewed and approved by the environmental protection authority according to the environmental standards during the business period. On the other hand, the livestock/poultry sectors must contribute a significant share to anthropogenic emissions of greenhouse gases (especially in methane) [21,22]. In this regard, the central governing authorities (i.e., the EPA and the COA) promoted the installation of the AD system in large-scale swine farms based on the dual benefits of environmental quality improvement and biogas-to-power for mitigating greenhouse gas (GHG) emissions. In order to promote the utilization of biogas from the swine-raising wastewater treatment for generating power and meeting the effluent standards (described in Section 3.2), Table 4 lists the rewards and subsidies based on the heads on farms in 2020. It should be noted that the rewards and subsidies shall not exceed 50% of their installation or acquisition costs for each item.

**Table 4.** The rewards and subsidies for the biogas-to-power and related facilities by the Council of Agriculture (COA) in 2020.

| Head on Farm | Reward/Subsidy Item | Upper Limit of Reward/Subsidy ($10^4$ NT$/Farm) [a] | Maximal Reward/Subsidy ($10^4$ NT$/Farm) |
|---|---|---|---|
| 2000~4999 | Biogas-to-power (installation reward) | 30 | 170 |
| | High bed facility | 25 | |
| | Rain/wastewater segregation system | 10 | |
| | Related biogas-to-power facilities | 75 | |
| 5000~9999 | Biogas-to-power (installation reward) | 60 | 270 |
| | High bed facility | 100 | |
| | Rain/wastewater segregation system | 15 | |
| | Related biogas-to-power facilities | 125 | |
| 10,000~14,999 | Biogas-to-power (installation reward) | 90 | 385 |
| | High bed facility | 125 | |
| | Rain/wastewater segregation system | 25 | |
| | Related biogas-to-power facilities | 175 | |
| ≥15,000 | Biogas-to-power (installation reward) | 120 | 500 |
| | High bed facility | 150 | |
| | Rain/wastewater segregation system | 35 | |
| | Related biogas-to-power facilities | 225 | |
| ≥15,000 (Fiduciary treatment) | Biogas-to-power (installation reward) | 175 | 600 |
| | High bed facility | 150 | |
| | Rain/wastewater segregation system | 35 | |
| | Related biogas-to-power facilities | 275 | |

[a] 1 US$ ≈ 30 NTD$.

### 3.1.3. Organic Agriculture Promotion Act

The Organic Agriculture Promotion Act, passed on 30 May 2018, was enacted to maintain water and soil resources, ecological environment, biodiversity, animal welfare, and consumer rights, as well as promoting an agricultural operation that is eco-friendly and a sustainable use of resources. The term organic agriculture was defined as any farming practice including cultivation, forestry, aquaculture, and animal husbandry without using chemical fertilizer, chemical pesticide, genetically modified organism (GMO), and related products, based on the principle of ecological balance and nutrient recycling. On the other hand, organic agriculture may contribute to renewable energy supply from the AD processes, thus fewer greenhouse gas emissions. As mentioned above, digestate is a by-product from the AD of the biogas plant production, which is rich in in minerals (such as nitrogen, phosphorus, calcium, and magnesium) and organic matter [14]. In this regard, biogas-digestate has been used as a sustainable fertilizer for crop production in the organic farming system (OFS) [17–20]. More significantly, the organic agriculture promotion in Taiwan echoed the UN Agenda 2030 for the SDGs. As compared to the organic agriculture area of about 13,500 hectares in 2019 [6], the SDG, in 2030, has been set up to 30,000 hectares in the Target 12.8.1 ("Area for environment-friendly and organic agriculture promotion"), which was based on Taiwan's Sustainable Development Goals announced by the central government in July 2019.

### 3.2. Environmental Protection Administration (EPA)

### 3.2.1. Water Pollution Control Act

The act, recently revised in June 2018, was formulated to control water pollution and ensure water quality for maintaining ecological systems and living environment. It is well known that the swine-raising wastewater contains considerable amounts of dissolved organic matters in terms of chemical oxygen demand (COD) and other dissolved pollutants. In this regard, the relevant measures for controlling the swine-raising discharges include the following items:

- Those swine-raising enterprises that discharge wastewater or sewage into surface water bodies shall comply with the effluent standards, which will be announced by the central competent authority;
- The residual sludge produced by the swine-raising wastewater or sewage treatment processes shall be properly managed and not be arbitrarily stored and dumped;
- The central competent authority shall collect water pollution control fees from the swine-raising enterprises based on their discharge water volume and water quality (e.g., chemical oxygen demand and suspended solid);
- Prior to the establishment, the swine-raising enterprises must submit the water pollution control plan and related documents for review and approval by the local governments;
- Those swine-raising enterprises that discharge wastewater or sewage into surface water bodies must apply to the local governments for discharge permit.

Under the authorization of the Water Pollution Control Act, Table 5 lists the main effluent standards for livestock/poultry-raising farms, showing that the standard item for COD must be below 600 mg/L for the swine-raising farms. It should be noted that the swine-raising owners may be claimed to pay the water pollution control fees under the authorization of the Water Pollution Control Act. The fees for the swine-raising farms were calculated according to the water quality items of treated wastewater (effluent) by the following rates: 12.5 NTD$/kg COD, 0.62 NTD$/kg suspended solid (SS), 625 NTD$/kg copper (Cu), and 625 NTD$/kg zinc (Zn).

In order to resolve the problem and valorize the value-added resources from the anaerobic fermentation of swine-raising manure, the central competent authority simplified the application and review procedures according to the "Water Pollution Control Measures and Test Reporting Management Regulations" on 27 December 2017. More significantly, the swine-raising farms can be

exempted from paying water pollution control fees if they shall take any of the following measures in the treatment of swine manure for recycled use in compliance with the related regulations:

- The competent authority of agriculture approved the recycled reuse of swine manure for the irrigation of farmlands based on the Waste Management Act (described thereafter);
- The competent authority of agriculture approved the recycled reuse of liquor and biogas digestate from the anaerobic digestion process as organic fertilizers for farmlands;
- The local governments approved the recycled use of treated wastewaters from the swine-raising farms for irrigation of plantation lands, which meet the effluent standards (see Table 4).

In addition, the central competent authority also requested swine-raising farms registered after the amendments to recycle at least 10% of their discharged wastewaters. The existing swine-raising farms with the establishment prior to 27 December 2017, those that discharged wastewater to the surface water body, shall meet the recycling ratios as follows:

- Large-sized swine-raising farms with more than 2000 heads will meet at least 5% of recycling wastewater generated in 5 years (by the end of 2022), and 10% of recycling wastewater generated in 10 years (by the end of 2027) from 27 December 2017;
- The medium-sized swine-raising farms with 20–2000 heads will meet at least 5% of recycling wastewater generated in 8 years (by the end of 2025), and 10% of recycling wastewater generated in 12 years (by the end of 2029) from 27 December 2017.

**Table 5.** Main effluent standards for livestock/poultry-raising farms.

| Item | Limit | Remark |
|---|---|---|
| pH | 6.0–9.0 | |
| Nitrate nitrogen | 15 mg/L | Not applicable to total nitrogen control. |
| Ammonia nitrogen | 50 mg/L | 1. Applicable to discharge into tap water quality and volume protection area. 2. The COA will announce the control date and effluent standard in consultation with the EPA. |
| Copper (Cu) | 3 mg/L | |
| Zinc (Zn) | 5 mg/L | |
| Biochemical oxygen demand (BOD) | 80 mg/L | |
| Chemical oxygen demand (COD) | 600/450 mg/L | The limits of 600 and 450 mg/L are applicable to the non-herbivorous animals (i.e., pig, chicken, duck, and goose) and herbivorous animals (i.e., cattle, horse, sheep, deer, and rabbit), respectively. |
| Suspended solids (SS) | 150 mg/L | |

3.2.2. Waste Management Act

The aim of this act, which was recently revised on 14 June 2017, was to effectively manage waste for maintaining environmental quality and public health. In order to promote the circular economy, the reuse of industrial waste as available resource shall be processed in accordance with the regulations, which were stipulated by the central industry competent authorities under the authorization of this act. In this regard, the central competent authority of agriculture has announced that the swine manure or manure from solid–liquid separation can be reused as raw material for the production of organic fertilizer, cultivation medium, biomass energy, or fuel. On the other hand, the products derived from the specified reuse methods should be in compliance with the national/international standards, or related regulations like the Fertilizer Management Act and the Renewable Energy Development Act (described thereafter).

### 3.2.3. Soil and Groundwater Pollution Remediation Act

The act, recently amended in February 2010, was to prevent and remediate soil and groundwater pollution. Regarding the reuse of liquor and biogas digestate as organic fertilizers for farmlands, the local governments shall conduct a regional monitoring of quality of liquor and biogas digestate, soil, and groundwater every two years after its approval. The required items to be monitored include the following:

- Quality of liquor and biogas digestate by monitoring pH, conductivity, total nitrogen, total phosphorous, copper, zinc, and so on;
- Quality of groundwater by monitoring conductivity, ammonia nitrogen, and so on;
- Quality of soil of fertilized farmland by monitoring conductivity (soil saturated extract), copper, zinc, and so on.

### 3.3. Ministry of Economic Affairs (MOEA)

In Taiwan, the central legislation for promoting renewable energy and establishing a green energy industry was built on the Renewable Energy Development Act (REDA), which was first passed on 8 July 2009 and recently revised on 1 May 2019. Obviously, the biogas-to-power is one of the renewable energy sources because it was derived from the anaerobic digestion of domestic organic waste. Under the authorization of the REDA, the feed-in tariffs (FIT) for the biogas-to-power in Taiwan have been announced annually [23–25], indicating an upward trend from 2.0615 in 2010 to 5.1176 NTD\$/kW-h (1 US\$ $\approx$ 30 NTD\$) in 2020. As mentioned above, the installation of a biogas-to-power system by the swine-raising owners can be also subsidized by the COA. These inter-ministry measures will provide promotional incentives for installing a biogas-to-power system by the swine-raising owners. Based on the FIT rate (5.1176 NTD\$/kW-h), biogas generation per head (about 0.1 m$^3$/day), and biogas power efficiency (about 1.4 kW-h/m$^3$), the income by selling biogas-to-power can get approximately $5.23 \times 10^8$ NTD\$ from the swine-raising industry with $200 \times 10^4$ heads.

### 4. Conclusions and Prospective

In Taiwan, the number of swine farms totaled 6645, raising a total of about 5.5 million heads with valuing over one billion US\$ per year. However, the swine-raising industry is facing many problems. Among them, the swine-raising wastewater treatment and its derived residues in compliance with the environmental regulations was probably the biggest problem to be resolved. On the other hand, the livestock sector significantly contributes to the release of greenhouse gas emission (i.e., methane). In this regard, the AD process for treating livestock manure has been proven to be a cost-effective valorization technology. Over the past years (2017–2019), Taiwan's Environmental Protection Administration, jointly ventured with the Council of Agriculture, announced the promotional measures on the reuse of liquor and biogas digestate from the anaerobic digestion of swine manure for farmland fertilization. It showed that the goals for promoting an irrigation area have been achieved from 584 hectares in 2017 to 2645 hectares in 2019. In 2019, the carbon dioxide emission reduction was equivalent to about 36,000 metric tons based on a total of 2.35 million metric tons liquor and biogas digestate applied and 2 million swine heads for the biogas-to-power. This innovative process of valorizing the value-added resources from the swine-raising manure management in Taiwan provided several benefits, including biomass energy production, greenhouse gas reduction, environmental quality improvement, and organic agriculture promotion.

To raise the biogas-to-power installation in the small and medium swine operators, some centralized AD units shall be built at proper sites due to the cost considerations. Under the authorization of the Renewable Energy Development Act, Taiwan's MOEA shall adopt further promotional FIT rates for the small and medium swine operators, which installed the centralized AD processes for generating biogas. In addition, the swine-raising enterprises (at least 5000 heads on farm) may expand their economic scale through mergers.

**Author Contributions:** Conceptualization, Y.-R.L.; data collection, Y.-R.L.; data analysis, W.-T.T.; writing—original draft preparation, W.-T.T.; writing—review and editing, W.-T.T. All authors have read and agreed to the published version of the manuscript.

**Funding:** This research received no external funding.

**Conflicts of Interest:** The authors declare no conflict of interest.

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
