# Peer review of "Valorization of Value-Added Resources from the Anaerobic Digestion of Swine-Raising Manure for Circular Economy in Taiwan"

_fermentation, doi:10.3390/fermentation6030081_

Round 1
Reviewer 1 Report
This is an interesting study of the regulations surrounding the use of anaerobic digestion products as part of the circular economy in order to reduce carbon dioxide emissions.
General Notes:
-Edit language for proper pronoun agreement and a ensuring a clear antecedent is present
-Acronyms are introduced multiple times, and some of the acronyms appear upon the second or third usage of the organization name. Several are given in the abstract as well as in the body of the report, would recommend only giving in the main text. As well, if the acronym is not used following it’s introduction, it does not need to be used.
-Authors discuss the use of biogas digestate as an organic fertilizer at several points. Line 88 states that the effluent of anaerobic digestion meets the standards, but the authors do not state if the digestate meets the requirements for an organic fertilizer as outlined in Table 3. More detail specifically showing where and how the digestate does/does not meet the standards set by the multiple governing bodies would be a valuable addition to this piece.
Specific points:
-Line 40 – Did the chapter added introduce legislation for the digestate and liquor reuse that contributed to the meeting the goals or was it only suggestions for the industry to follow?
-Line 45 and 288 - the word 'irritation' is used. Is this to mean irrigation? Or is this a field specific term? Also, the two sentences in these lines (one in the introduction and one in the conclusion) very closely mirror each other despite being in very different sections.
-Line 74 – typo for Therefore
-Line 74 – is there a reference or a source for the claim that the effluent standards were responsible for the closure of smaller scaled farms? In the previous line, authors attribute increased global competition as the cause for the loss of 1.3 million heads, would that not also be a cause for smaller farms to close?
Line 85 – missing adjective ‘to’ before ‘water bodies’
Line 112 – Authors write ‘United States’ by accident when referring to the United Nations
Line 135 – dates should be written ‘june 18, 1999’ and ‘June 19, 2002’
Line 160 – ‘play’ is used when possibly meaning ‘plan’
Author Response
Q1. Edit language for proper pronoun agreement and an ensuring a clear antecedent is present.
Reply: As suggested by the reviewer, the English expressions have been corrected in the revised version of the manuscript.
Q2. Acronyms are introduced multiple times, and some of the acronyms appear upon the second or third usage of the organization name. Several are given in the abstract as well as in the body of the report, would recommend only giving in the main text. As well, if the acronym is not used following it’s introduction, it does not need to be used.
Reply: As suggested by the reviewer, the use of acronym in academic writing will be followed by the comments.
Q3. Authors discuss the use of biogas digestate as an organic fertilizer at several points. Line 88 states that the effluent of anaerobic digestion meets the standards, but the authors do not state if the digestate meets the requirements for an organic fertilizer as outlined in Table 3. More detail specifically showing where and how the digestate does/does not meet the standards set by the multiple governing bodies would be a valuable addition to this piece.
Reply: As suggested by the reviewer, the relevant descriptions have been added to the last paragraph of section 2.2. Also, Fig. 1 has been corrected to make it clear.
“….. In addition, the liquor treated by the AD process can be reused for irrigation of plantation land if it meets the effluent standards and was approval by the local competent authorities. If the digestate does not meet the requirements for an organic fertilizer, it will be handled by the waste management regulations.”
Q4. Line 40 – Did the chapter added introduce legislation for the digestate and liquor reuse that contributed to the meeting the goals or was it only suggestions for the industry to follow?
Reply: As suggested by the reviewer, the background about the regulation has been added to provide some environmental goals and contributions.
“To reach environmental goals in connection with livestock industry, such as improving river quality, air quality, and circular economy objectives, the Taiwan’s Environmental Protection Administration, jointly ventured with the Council of Agriculture, announced the amendments to the “Water Pollution Control Measures and Test Reporting Management Regulations” under the authorization of the Water Pollution Control Act on November 24, 2015. A featured chapter on the reuse of liquor and biogas digestate from the anaerobic digestion of swine manure for farmland fertilization has been added to the Regulation.”
Q5. Line 45 and 288 - the word 'irritation' is used. Is this to mean irrigation? Or is this a field specific term? Also, the two sentences in these lines (one in the introduction and one in the conclusion) very closely mirror each other despite being in very different sections.
Reply: The word ‘irrigation’ should be used. So, this typo has been corrected in the manuscript.
Q6. Line 74 – typo for Therefore
Reply: This typo has been corrected in the manuscript.
Q7. Line 74 – is there a reference or a source for the claim that the effluent standards were responsible for the closure of smaller scaled farms? In the previous line, authors attribute increased global competition as the cause for the loss of 1.3 million heads, would that not also be a cause for smaller farms to close?
Reply: As suggested by the reviewer, an additional reference (Ref. 10) has been added to the manuscript.
Q8. Line 85 – missing adjective ‘to’ before ‘water bodies’
Reply: This sentence has been corrected by adding “into” before ‘water bodies’.
Q9. Line 112 – Authors write ‘United States’ by accident when referring to the United Nations
Reply: This typo has been corrected in the manuscript.
Q10. Line 135 – dates should be written ‘June 18, 1999’ and ‘June 19, 2002’
Reply: These dates have been corrected in the manuscript.
Q11. Line 160 – ‘play’ is used when possibly meaning ‘plan’
Reply: This typo has been corrected in the manuscript.

Reviewer 2 Report
The manuscript can benefit by addressing the following issues:
- The writing (grammar and sentence constructions) of the manuscript must be improved. There are several misspelled words throughout the manuscript (e.g., therfore in line 74). Some acronym are not accurate as well (e.g., United States (UN) in line 112)
- The title is misleading, specifically the term "environmental sustainability", which supposed to include all the environmental benefit of the process. However, it seems that the only focus of the report was on GHG.
- The environmental impact (on water, soil, air) of the case study should have been compared to a base case scenario (e.g., anaerobic digestion was not applied to swine manure). Doing so would provide a measure of how sustainable or how environmentally-friendly the process is.
- The study was focused mainly on large swine-raising operations. The authors should provide some perspective or recommendations on how to economically apply the process to small operators (e.g., centralized anaerobic digestion unit).
Author Response
Q1. The writing (grammar and sentence constructions) of the manuscript must be improved. There are several misspelled words throughout the manuscript (e.g., therefore in line 74). Some acronym are not accurate as well (e.g., United States (UN) in line 112).
Reply: As suggested by the reviewer, the English expressions and typo errors have been corrected in the revised version of the manuscript.
Q2. The title is misleading, specifically the term "environmental sustainability", which supposed to include all the environmental benefit of the process. However, it seems that the only focus of the report was on GHG.
Reply: Indeed, the title seemed out of focus on environmental sustainability. However, the main objectives of this case report included some indicators and goals in the Taiwan’s sustainable development goals with relevance to livestock industry, which are based on the United Nations sustainable development goals (SDGs).
Q3. The environmental impact (on water, soil, air) of the case study should have been compared to a base case scenario (e.g., anaerobic digestion was not applied to swine manure). Doing so would provide a measure of how sustainable or how environmentally-friendly the process is.
Reply: As suggested by the reviewer, the description about the environmental impacts of reusing the biogas digestate & liquor from the anaerobic digestion of swine-raising manure has been added to the Section 3.
“As mentioned above, the issues for the valorization of value-added resources (i.e., biogas digestate and biogas-to-power) from the anaerobic fermentation of swine manure are related to the central competent authorities, including the COA, the EPA and the MOEA. Using AD liquor and solid digestate as agricultural fertilizer and other reuse purposes, instead of flushing livestock slurry away, thus reduces water and electricity costs for the enterprises concerned. Livestock slurry is also diverted from polluting receiving bodies in the Taiwan’s rivers. The information about the regulations for promoting the value-added resources from the swine manure will be addressed in the following sections by using the official regulatory website [12].”
Q4. The study was focused mainly on large swine-raising operations. The authors should provide some perspective or recommendations on how to economically apply the process to small operators (e.g., centralized anaerobic digestion unit).
Reply: As suggested by the reviewer, some perspectives (e.g., centralized anaerobic digestion unit) have been addressed to economically apply the process to small swine-raising operators.
“To raise the biogas-to-power installation in the small and medium swine operators, some centralized AD units shall be built at proper sites due to the cost considerations. Under the authorization of the Renewable Energy Development Act, the Taiwan MOEA shall adopt further promotional FIT rates for the small and medium swine operators, which installed the centralized AD processes for generating biogas. In addition, the swine-raising enterprises (at least 5,000 heads on farm) may expand their economic scale through mergers.”

Round 2
Reviewer 2 Report
Q2. The title is misleading, specifically the term "environmental sustainability", which supposed to include all the environmental benefit of the process. However, it seems that the only focus of the report was on GHG.
Reply: Indeed, the title seemed out of focus on environmental sustainability. However, the main objectives of this case report included some indicators and goals in the Taiwan’s sustainable development goals with relevance to livestock industry, which are based on the United Nations sustainable development goals (SDGs).
- I still think that the Title has to be revised to reflect the content of the manuscript.
Author Response
Q1. I still think that the Title has to be revised to reflect the content of the manuscript.
Reply: As suggested by the reviewer, the title has been changed to be relevant to the content of the manuscript for circular economy benefits.
